# Topical Wound Oxygen Therapy in the Treatment of Chronic Diabetic Foot Ulcers

**DOI:** 10.3390/medicina57090917

**Published:** 2021-08-31

**Authors:** Robert G. Frykberg

**Affiliations:** Diabetic Foot Consultants, LLC, Midwestern University, Glendale, AZ 85308, USA; rgfdpm@diabeticfoot.net

**Keywords:** oxygen, topical oxygen therapy, diabetic foot ulcers, wound healing

## Abstract

Oxygen is a critical component of many biological processes and is essential for wound healing. Chronic wounds are typically characterized as being hypoxic in that the partial pressure of oxygen (pO_2_) in the center of the wound is often below a critical threshold necessary to fully support those enzymatic processes necessary for tissue repair. Providing supplemental oxygen can effectively raise pO_2_ levels to better optimize functioning of these essential enzymes. While hyperbaric oxygen therapy has been well studied in this regard, comparative clinical studies have fallen short of providing clear evidence in support of this modality for healing chronic diabetic foot ulcers (DFU). Topical oxygen therapy (TOT) has been in clinical use for over 50 years with encouraging pre-clinical and clinical studies that have shown improved healing rates when compared to standard care. Nonetheless, TOT has heretofore been discounted as an unproven wound healing modality without theoretical or clinical evidence to support its use. This review shall provide a brief summary of the role of oxygen in wound healing and, specifically, discuss the different types of topical oxygen devices and associated studies that have convincingly shown their efficacy in healing chronic DFUs. The time has come for topical oxygen therapy to be embraced as a proven adjunctive modality in this regard.

## 1. Introduction

While many clinicians might consider topical oxygen therapy (TOT) to be an unproven or controversial wound healing modality, it has been in clinical use for over fifty years. In his 1969 publication, Fischer described his novel topical “hyperbaric” oxygen system used to treat a variety of chronic wounds in an in-patient environment [1]. Using humidified oxygen under a constant pressure of 22 mmHg for 4–12 h per day, he was able to achieve success in 88% of his cases including diabetic foot ulcers (DFU), venous leg ulcers (VLU), and various decubitus pressure ulcers. In a subset of six patients with bilateral lesions using one side as a control, only the six “hyperbaric oxygen” treated wounds healed within 3 to 17 days. When the unhealed control wounds were subsequently switched to topical oxygen therapy, they all healed within 6 weeks. While this rudimentary case series was not up to the scientific standards of present-day clinical investigations, it was certainly compelling enough to lead to further applications for topically applied oxygen therapies.

Most unfortunately, almost 20 years later, the misleading TOT clinical trial by Leslie et al. was published in 1988 [2]. Often cited as the “evidence” to prove that topical oxygen therapy is not effective in healing DFU (or any other chronic wound), this clinical trial was a distortion of what might be considered a reasonable randomized controlled trial. The investigators enrolled only 28 patients, administered the topical “hyperbaric” oxygen (THO) therapy for only 2 weeks, and assessed primary healing outcomes of the DFU at 14 days. To expect to test the efficacy of any therapy within a short two-week period in a very underpowered study is, in itself, an exercise in futility. Although both groups, as compared to baseline, had significant reductions in wound area and depth by day 14, there were no differences between the two groups for these parameters. The authors therefore concluded that healing of DFU was not accelerated by THO in their study. Sadly, this very poorly designed study led to many misconceptions about the utility of topical oxygen therapy for managing chronic wounds for several decades.

Systemically administered hyperbaric oxygen therapy (HBOT), in contrast to TOT, has been well studied and embraced by the wound healing communities world-wide [3]. Although the putative physiologic benefits of HBOT are indeed compelling, clinical trials aimed at supporting its efficacy in healing chronic DFU have been contradictory and disappointing [4,5,6]. 

This review aims to briefly explore the role of oxygen in the healing wound and, specifically, to address the evidence supporting the benefits and clinical efficacy of TOT for healing chronic diabetic foot ulcers. 

### Oxygen and Wound Healing

Recognizing that Oxygen (O_2_) is required for almost every step of the response to the injury and wound healing cascade, several recent reviews have focused not only on the role of molecular oxygen in this regard but also on cellular and biochemical mechanisms for O_2_ generation [3,7,8,9,10]. Chronic wounds are typically characterized as being hypoxic in that the partial pressure of oxygen (pO_2_) in the center of the wound is often below a critical threshold necessary to fully support those enzymatic processes necessary to regenerate tissue [10]. Disrupted vascular supply, chronic inflammation, bacterial overload, and exhausted local metabolic oxygen production all contribute to chronic hypoxia. While acute, short-term hypoxia can indeed be a stimulus for angiogenesis, chronic hypoxia impedes not only angiogenesis but also the associated generation of reactive oxygen species (ROS) necessary for the upregulation of growth factors, cell signaling, and bacterial killing [3]. Oxygen is the rate limiting substrate for numerous biochemical reactions and plays a crucial role in energy production and cellular metabolism. Molecular oxygen is, of course, also necessary for the synthesis of nitric oxide (NO) that regulates vasodilatation. Oxygen-dependent processes, so relevant in wound healing, include mitochondrial-driven adenosine triphosphate (ATP) production for chemical/cellular energy and nicotinamide adenine dinucleotide phosphate (NADPH) oxidase for the production of ROS (“respiratory burst”) involved in signal transduction of growth factors, cellular recruitment, and bacterial killing [8,9,10,11]. The two most prevalent ROS, superoxide and hydrogen peroxide (H_2_O_2_), both serve to upregulate the release of vascular endothelial growth factor (VEGF) and platelet derived growth factor (PDGF) that stimulate endothelial cell division and migration to initiate angiogenesis, lymphocyte/leukocyte migration, and fibroblast division and migration to synthesize new extracellular matrix (ECM). ROS driven phagocytosis and bacterial killing by bacteriostatic H_2_O_2_ release by platelets and neutrophils also play an important role in the initial clearing of bacterial pathogens [9] (Table 1).

Collagen synthesis, deposition, and polymerization can only take place provided there is the necessary molecular oxygen present for the hydroxylation of proline and lysine. Prolyl hydroxylase and lysyl hydroxylase are oxygen dependent and very sensitive to local pO_2_ levels, with maximal activity at oxygen levels approaching 250 mmHg [8,10]. Since the pO_2_ in the center of a chronic wound can be as low as 10 mmHg, maximal enzymatic activity and collagen production can only be achieved by increasing oxygen concentrations above normal physiologic levels. In fact, this is also true for NADPH oxidase where maximal ROS production occurs at oxygen levels approaching 300 mmHg [8,12]. Supplemental oxygen should therefore augment all of the biological processes necessary for healing by raising the pO_2_ above those levels found not only in chronic wounds but also above normal tissue levels of 100 mmHg. The question remains, however, is there sufficient evidence to support a positive tissue response from topically applied oxygen? 

## 2. Topical Oxygen Devices

Topical oxygen therapy (TOT) can be defined as the administration of oxygen applied topically over injured tissue by either continuous diffusion or pressurized systems. Although there are dressings, gels, and hemoglobin sprays that can provide for oxygen release when applied to wounds, for the purposes of this discussion only mechanical devices specifically indicated for topical oxygen therapy will be discussed herein [3].

There are three general types of physical delivery systems for TOT: (1) those providing for continuous delivery of oxygen (CDO) (TransCu O_2_^®^ (EO2 Concepts); EpiFlo^®^ (Ogenix); and Natrox^®^ (Inotec), (2) those providing low constant pressure in a contained chamber (GRW Medical (Chadds Ford, PA, USA), OxyCare^®^ (Bremen, Germany) and (3) those that are cyclically pressurized and humidified in a contained chamber [TWO2^®^ (AOTI, Galway, Ireland) (Figure 1). 

CDO devices apply topical continuous diffusion of non-pressurized (normobaric) pure oxygen through small cannulas or thin tubes to semi-occlusive or proprietary wound dressings. Small portable, battery powered, electrochemical oxygen generators supply a continuous flow of pure oxygen over the wounds 24 h per day at a flow rate of up to 15 mL/h [13,14,15]. An oxygen gradient then develops between the overlying dressing and the wound bed, thereby facilitating oxygen diffusion. The wound dressings are typically changed weekly, and the oxygen generators (or batteries) are generally replaced after 1 to 2 weeks of continuous use. These light-weight devices can be held in a small pouch affixed to the patient’s leg or hip and allow for unrestricted ambulation within the prescribed offloading devices. There have been several recently published randomized controlled trials (RCT) that attest to their ease of use and positive effect on DFU wound healing [15,16,17,18].

The lower constant pressure devices provide oxygen delivery in a simple plastic boot that is placed over the extremity with the ulcer. One hundred percent oxygen is delivered for 90 min for 4 consecutive days per week. Constant pressure is then maintained within the chamber up to 22 mmHg (1.03 atm). Although less widely used than the other modalities, numerous studies have been conducted on these types of devices over the last four decades that have shown good clinical efficacy. However, the majority of these studies have consisted of case series or uncontrolled trials, including one animal study [19,20,21]. The one very poorly conducted RCT that used a similar device has been previously discussed [2]. A more recent retrospective chart review of a variety of non-healing wounds in patients from the manufacturer’s database reported that >50% of wounds less than 1 year in duration experienced healing [22].

The Topical Wound Oxygen (TWO_2_) system differs from other devices in that it applies cyclically pressurized (10–50 mb) pure oxygen within a disposable extremity chamber connected to a stationary oxygen concentrator. Humidity can be added to the system if required. The benefit of this approach is that the higher pressure gradient (pO_2_) results in oxygen molecules diffusing deeper into the hypoxic wound tissue to enhance multiple molecular and enzymatic functions [20,23]. Within the extremity chamber containing pure O_2_ at sea level (760 mmHg), the pO_2_ can be cyclically pressurized up to nearly 800 mmHg that optimizes enzymatic activity as previously discussed [8]. The cyclical pressure applied with TWO_2_ of between 8 mmHg and 38 mmHg creates sequential non-contact compression of the limb that also helps to reduce peripheral edema, and thereby, stimulate wound site perfusion further [24,25]. Several prospective clinical studies have been successfully conducted using this device on both VLUs as well as DFUs [25,26,27].

## 3. Topical Oxygen Effect on Wound Healing—The Evidence

In the last several decades there have been several case series and reviews suggesting the putative benefits of topical oxygen administration to expedite healing of chronic wounds [7,11,19,21,28,29,30]. Perhaps the most compelling early evidence for a topical oxygen wound healing signal was presented in the pre-clinical animal study by Fries et al. published in 2005 [20]. A series of excisional dermal wounds in pigs were created; half of the wounds were subjected to topical oxygen therapy whereas matched control wounds were exposed to ambient air. In one set of experiments, an O_2_ electrode was placed at a 2 mm depth in the center of the wound bed. After an exposure of just 4 min, the central superficial wound pO_2_ of 11 mmHg increased four-fold to levels of 40 mmHg. Topical O_2_ was administered for 7 consecutive days and then stopped. Wound site pO_2_ was then measured on day 22, showing a sustained and significantly higher pO_2_ in the treated wounds compared to controls. Repeated treatment accelerated wound closure. Histological studies revealed that oxygen-treated wounds showed a stronger presence of VEGF at 7 days, signs of improved angiogenesis at 16 days, and, at day 22, advanced tissue remodeling with normal architecture and healing in the treated group compared to control wounds on the same animals [20]. In a subsequent human study, this same group found that TOT significantly improved wound healing and, in these wounds, there was a significantly higher VEGF expression compared to non-healing wounds [19]. Interestingly, they did not find a significant increase in VEGF expression in those patients treated with HBOT. Similar increases in VEGF and other cytokines in healers vs. non-healers were reported more recently in patients treated with CDO [14,31].

An increasing number of prospective (as well as retrospective), comparative clinical studies have provided the translational evidence necessary to support the efficacy of TOT in conjunction with the standard of care for healing chronic wounds [7]. Although most clinical research has focused on DFUs, several prospective, comparative cohort studies have also shown significantly improved healing of VLUs using cyclical pressurized TWO_2_. One non-randomized study of 83 VLU patients measured the effect of TWO_2_ compared to conventional compression dressings (CCD) [24]. At 12 weeks, 80% of TWO_2_ managed ulcers were completely healed compared to 35% of the CCD managed ulcers. These same authors later conducted another VLU non-randomized comparative study that similarly investigated the efficacy of TWO_2_ vs. CCD in the management of refractory non-healing venous ulcers (RVU) with a duration of at least two years [25]. At 12 weeks, 76% of the TWO_2_ managed ulcers had completely healed, compared to 46% of the CCD-managed ulcers with a median time to full healing of 57 days and 107 days, respectively. No other formal VLU studies using TOT have been published to date.

In 2010, a small, prospective, non-blinded, non-randomized study was conducted to examine the clinical efficacy of topical wound oxygen therapy in healing ambulatory DFU patients [26]. Patients were simply allocated to the topical oxygen if a unit (TWO_2_) was available or were otherwise treated with advanced moist wound therapy. At 12 weeks 82.4% of the ulcers in the active therapy arm and 45.5% in the control standard of care arm had healed completely (*p* = 0.04). The median time to complete healing was 56 days in the TWO_2_ therapy arm and 93 days in the control standard of care arm (*p* = 0.0013).

A very small, preliminary single center RCT of 17 DFU patients, comparing those treated with a CDO device plus standard of care (SOC) with those treated with standard of care alone, was published in 2013 [17]. This study, where patients were followed for only four weeks, indicated that the topical oxygen group achieved an average wound size reduction of 87% compared to 46% in the standard of care group (*p* < 0.05). While significant wound sample macrophage and inflammatory cytokine level changes (IL-6, IL-8, MMP-1, and MMP-2) were noted in the active CDO group, these patients were not followed until complete healing and the sample size was too small to be widely generalizable [17].

Several years later, the results of the formal sham controlled, multicenter RCT using this same CDO device (EpiFlo) on University of Texas (UT) 1A category DFUs was published [16]. For the primary endpoint of complete healing at 12 weeks in the intention-to-treat (ITT) population (*n* = 128), 53.8% of active CDO patients healed compared to 49.2% of the control sham plus SOC patients (*p* = 0.42). While otherwise a generally well conducted trial, per protocol and other ad hoc sub-analyses also failed to show significantly improved healing rates over those seen in the SOC group [16].

In contrast, the pivotal trial of another CDO device (TransCu O_2_) reported significantly positive results in their 12-week, multicenter, blinded, placebo (sham) controlled, parallel group clinical trial [18]. Enrolling patients with UT class 1A ulcers of at least 30 days duration, patients first went through a two week run-in period of standard care. Randomizing 146 patients considered for ITT analysis, the primary outcome again was the percent of patients in each group achieving complete healing at 12 weeks. They reported that 32.4% of CDO treated patients completely healed compared to 17.7% of sham control patients that healed (95% CI 1.05 to 3.59, *p* = 0.033). This was a relative performance improvement of 195% compared with the placebo arm. The time to ulcer closure was also shorter in patients who received CDO therapy (*p* = 0.015). Despite a large number of exclusions after initial randomization and nearly 30% dropout rate, their ITT and per protocol (PP) analyses clearly demonstrated that their topical oxygen device healed significantly more DFU at 12 weeks than those treated by standard care alone [18].

Serena et al. very recently reported on their multicenter RCT comparing another CDO device (Natrox) against good standard of care for the healing of chronic DFUs [15]. This 12 week study, open-label and unblinded, uniquely randomized 145 patients with chronic DFU to either SOC, using primarily a total contact cast (TCC), or to the active TOT group plus SOC/TCC. With a primary outcome of complete healing at 12 weeks and intention-to-treat analysis, 18/64 (28.1%) patients healed in the SOC group at 12 weeks compared with 36/81 (44.4%) in the SOC plus TOT group (*p* = 0.044). In the PP population (those completing ≥ 8 visits), there was a statistically significant reduction in wound area between the groups: the SOC group saw a mean reduction of 40% compared to the SOC plus TOT group mean reduction of 70% (*p* = 0.005). Partially conducted amid the coronavirus pandemic, this study had a 19% withdrawal rate. Nonetheless, this community-based study has demonstrated that TOT can lead to a statistically significant improvement in healing rates in patients with DFUs that are resistant to healing with optimal SOC alone [15].

Studying home-based, cyclical pressurized topical wound oxygen (TWO_2_) for the healing of recalcitrant DFUs, a rather robust multicenter, multinational, sham controlled, double-blinded RCT was recently reported by Frykberg et al. [27]. Using a group sequential design with a priori stopping points and optimal SOC throughout, the active TWO_2_ arm was found to be superior to the sham arm, with a closure rate of 41.7% at 12 weeks compared with 13.5% (*p* = 0.007), respectively. Favoring those treated with the active therapy, this difference in outcome produced an odds ratio (OR) of 4.57 (97.8% CI 1.19, 17.57, *p* = 0.010). Cox proportional hazards modeling, after adjustment for UT grade, demonstrated >4.5 times the likelihood to heal DFUs over 12 weeks compared with the sham arm with a hazard ratio (HR) of 4.66 (97.8% CI 1.36, 15.98, *p* = 0.004). Furthermore, they found that 56% of TWO_2_ patients achieved 100% healing at 12 months vs. 27% in the sham arm, (*p* = 0.013) and only a 5% amputation rate at one year from enrollment. As found in other TOT cohort and randomized trials, this sham controlled, double-blind RCT demonstrated that, at both 12 weeks and 12 months, adjunctive cyclical pressurized TWO_2_ therapy was superior in healing chronic DFUs compared with optimal SOC alone [27].

This same device has subsequently been investigated to examine the real-world impact of TWO_2_ on hospitalizations and amputations in patients with diabetic foot ulcers (DFU) compared to patients who had not used TWO_2_. An, as of yet, unpublished retrospective, comparative cohort study of 202 DFU patients found that 6.6% and 12.1% of TWO_2_ patients had hospitalizations and amputations at one year, respectively, compared to 54.1% and 41.4% of patients who had not used adjunctive TWO_2_ (*p* < 0.0001, *p* < 0.0001), representing 88% and 71% reductions [32]. Although this data is still subject to peer review, it infers that treating DFU patients with TWO_2_ can lead to significant reductions in hospitalizations and amputations in the real-world setting.

Two recent systematic reviews with meta-analyses have further added support to the clinical effectiveness of topical oxygen therapy for healing chronic DFUs [33,34]. Although not without some methodological deficiencies and heterogeneity, they both indicate that TOT (using CDO devices as well as cyclical pressurized devices) can augment wound healing among persons with chronic diabetic foot ulcers. Although the exact location of the wounds being treated in these studies was not specified, most were on the plantar surfaces. Hence, appropriate offloading was a critical component of standard of care in both assignment groups within each study. Thanigaimani et al., reviewing all but the most recent 2021 TOT trial, indicated that their meta-analysis also suggests that TOT improves the likelihood of DFU healing [34]. However, not having the data cited above on amputations nor cost-effectiveness data, they could not make further suggestions regarding these outcomes.

## 4. Conclusions

From the foregoing, it is evident that topical oxygen therapy can no longer be considered an experimental or unproven therapy for the healing of chronic wounds, especially diabetic foot ulcers. The data clearly have demonstrated a significant improvement in the healing of chronic DFUs treated with either CDO devices or pressurized devices (TWO_2_) as compared to standard of care alone. That being said, it is also critical to emphasize that TOT (as for any advanced wound therapy) must be administered in conjunction with optimal wound care. Without addressing the basic tenets of wound care (debridement, offloading, treatment of infection, treatment of ischemia, etc.), no therapy can be expected to miraculously heal a chronic wound. Furthermore, not all wounds are suitable for TOT and not all wounds thus treated will heal; this therapy is certainly not a panacea. However, when used *adjunctively* with optimal wound care, the aforementioned studies provide the clinical evidence necessary to support the use of topical oxygen in the management of chronic DFUs.

## Figures and Tables

**Figure 1 medicina-57-00917-f001:**
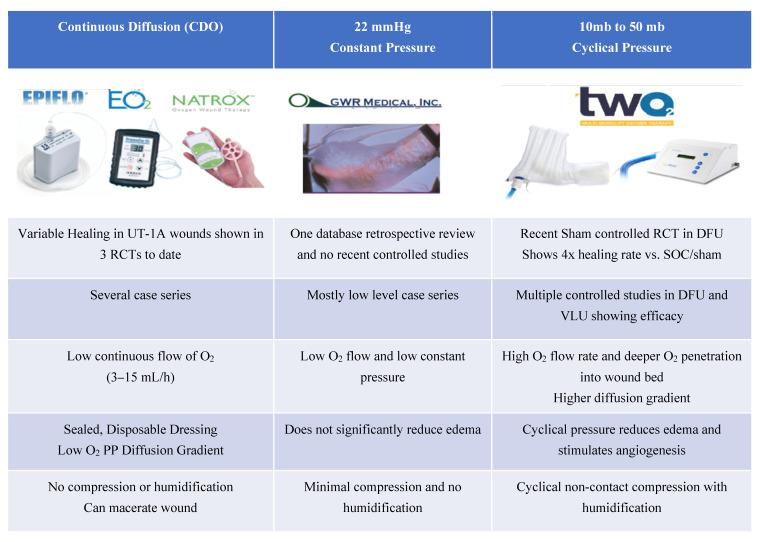
Types of topical oxygen devices.

**Table 1 medicina-57-00917-t001:** Role of Oxygen in Wound Healing.

Oxygen-Dependent Product	Enzyme or Substrate	Function	Cytokine, Cell Mediators; or Cellular/Tissue Effect
ATP	ATP synthase, Cytochrome C, Electronic Transport Chain	Chemical Energy for metabolism	
Reactive Oxygen Species (ROS) “respiratory burst” (Superoxide, Hydrogen peroxide (H_2_O_2_))	NADPH oxidase	Cellular Signaling/transduction Bacterial defenses Angiogenesis	Cell division and migration. Upregulation of Growth Factors (VEGF, PDGF, etc.) (leukocyte migration and phagocytosis, bacteriostatic H_2_O_2_) VEGF, PDGF, NO, etc.
Collagen synthesis	Prolyl hydroxylase, lysyl hydroxylase	Collagen deposition and crosslinking	Fibroblasts
Nitric oxide (NO)	Nitric oxide synthase	Vasodilatation, angiogenesis	Endothelium

NADPH: nicotinamide adenine dinucleotide phosphate; VEGF: vascular endothelial growth factor; PDGF: platelet derived growth factor.

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
