# Peer review of "Topical Wound Oxygen Therapy in the Treatment of Chronic Diabetic Foot Ulcers"

_medicina, 2021, doi:10.3390/medicina57090917_

Round 1

Reviewer 1 Report

Excellent review. Like with all reviews, the results are to be interpreted with caution, but the work is sound and as detailed as possible.

Author Response

Thank you for your review.

1. You mentioned "A weakness of the paper is the diversity of the included papers with their exact treatment methods."  I would think that a diversity included papers from different trial on different devices would actually be considered a strength in supporting the conclusion that TOT can augment healing of chronic DFU when used in conjunction with appropriate standard of care.

2.  You mentioned " It is unclear, whether the exact localization of the ulcer had an influence on the success of the oxygen treatment, or to a consequent off-loading of the affected zone and/or the climate in the closed chamber, which is necessary for the local oxygen treatment."  This is an astute observation, although not generally recorded as to specific locations of DFU in the randomized subjects (hence, through randomization and by definition, such sites should be fairly equally distributed). Nonetheless, the majority of wounds in all trials, as is common in practice, were on the plantar surface of the feet. To address the reviewer's concerns, I clarified this in the revision towards the bottom of page 2. As to the concerns regarding localized environments in chamber or under dressings, these parameters are integral to the TOT treatment modalities and as such cannot be further subdivided. As noted in the results of the two systematic reviews/meta-analyses, both types of CDO devices seemed to have a positive effect on healing compared to controls (with some differences of course in each modality). However, sham controls and dressings/chamber were identical to the active devices in terms of environments except for the application of pure oxygen rather than ambient air. Again, this contrasts active TOT therapy vs standard of care plus sham therapy between assignment groups.

3. Minor grammatical changes were made, including spacing errors. Thank you

Reviewer 2 Report

Very interesting article but I have few comments:

  1. Please, read clearly instructions for authors: "References should be numbered in order of appearance and indicated by a numeral or numerals in square brackets—e.g., [1] or [2,3], or [4–6]."
  2. Does the author have permission to use photos from the figure 1?
  3. Please complete the literature in the text (also in Tables).
  4. The author must standardize the markings, e.g. statistical significance.

Author Response

Thank you for your comments and allowing me to address them:

  1. Please, read clearly instructions for authors: "References should be numbered in order of appearance and indicated by a numeral or numerals in square brackets—e.g., [1] or [2,3], or [4–6]."      Although I thought this would be a simple formatting function prior to print, with some difficulty I added MDPI reference format to my Endnote program to change the printed formats of citations (not as easy as it should have been!)          Please also note: as is customary, all citations were ordered in the same sequence as appearing within the text, sometimes as grouped references.
  2. Does the author have permission to use photos from the figure 1?  NO, no attribution is necessary since this was just from my oral presentations
  3. Please complete the literature in the text (also in Tables).  I do not understand this comment. if this refers to citations, I have done so.
  4. The author must standardize the markings, e.g. statistical significance.         I have done this, although as the reviewer knows, different studies are reported differently in different journals and different comparisons are also reported variably (some as p<0.05 or p=0.005, or P=0.005). Where 95% CI was reported, I also included. However, not all hypothesis tests/comparisons are reported with confidence intervals.